# Preterm nutritional intake and MRI phenotype at term age: a prospective observational study

Vimal Vasu,[1,2] Giuliana Durighel,[3] Louise Thomas,[3] Christina Malamateniou,[4,5] Jimmy D Bell,[3] Mary A Rutherford,[4,5] Neena Modi[1]

For numbered affiliations see end of article.

**Correspondence to**
Dr Vimal Vasu;
vimal.vasu@nhs.net

## ABSTRACT

**Objective:** To describe (1) the relationship between nutrition and the preterm-at-term infant phenotype, (2) phenotypic differences between preterm-at-term infants and healthy term born infants and (3) relationships between somatic and brain MRI outcomes.

**Design:** Prospective observational study.

**Setting:** UK tertiary neonatal unit.

**Participants:** Preterm infants (<32 weeks gestation) (n=22) and healthy term infants (n=39)

**Main outcome measures:** Preterm nutrient intake; total and regional adipose tissue (AT) depot volumes; brain volume and proximal cerebral arterial vessel tortuosity (CAVT) in preterm infants and in term infants.

**Results:** Preterm nutrition was deficient in protein and high in carbohydrate and fat. Preterm nutrition was not related to AT volumes, brain volume or proximal CAVT score; a positive association was noted between human milk intake and proximal CAVT score (r=0.44, p=0.05). In comparison to term infants, preterm infants had increased total adiposity, comparable brain volumes and reduced proximal CAVT scores. There was a significant negative correlation between deep subcutaneous abdominal AT volume and brain volume in preterm infants (r=−0.58, p=0.01).

**Conclusions:** Though there are significant phenotypic differences between preterm infants at term and term infants, preterm macronutrient intake does not appear to be a determinant. Our preliminary data suggest that (1) human milk may exert a beneficial effect on cerebral arterial vessel tortuosity and (2) there is a negative correlation between adiposity and brain volume in preterm infants at term. Further work is warranted to see if our findings can be replicated and to understand the causal mechanisms.

### Strengths and limitations of this study

- There have been no previously published studies regarding the relationship between preterm nutritional intake and MRI outcomes at term age.
- This study provides comprehensive ascertainment of preterm nutritional data in parallel with somatic and brain MRI.
- Another strength of this study is the use of term born infants as comparator for MRI outcomes.
- Weaknesses included limited sample size and the prospective observational nature of the study.

deficiency remains a significant risk and our previously published data in 18 preterm infants from a single centre indicate that the preterm diet is low in protein while being high in carbohydrate and fat.[4] Our data also suggest that preterm macronutrition may affect later health by demonstrating a positive association between first week lipid intake in preterm infants and elevated intrahepatocellular lipid, which in adults is associated with the cardiometabolic syndrome.[5]

The preterm phenotype at term is characterised by aberrant adipose tissue (AT) partitioning,[6] reduced proximal cerebral arterial vessel tortuosity (CAVT),[7] reduced deep grey matter volumes[8] and reduced cerebral cortical folding.[9] The somatic phenotype observed is a matter of concern as adiposity is associated with inflammation and reduced brain volume in the adult population.[10]

Here, we present prospective observational data designed to (1) assess the influence of nutrition on the preterm phenotype at term age, (2) describe phenotypic differences between preterm infants at term and term healthy infants and (3) examine relationships between somatic and brain MRI measurements. The a priori hypotheses of our study were that (1) preterm macronutrient intake would be positively associated with central nervous system phenotype (brain

## INTRODUCTION

Preterm nutritional guidelines are based on consensus expert opinion[1][2] rather than compelling evidence and the 'optimal' diet for long-term health remains unknown.[3] Though there is now greater clinical emphasis on preterm nutrition, protein

volume and cerebral vessel tortuosity) and (2) preterm macronutrient intake would be negatively associated with internal abdominal (visceral) adiposity. The relationship between somatic and brain MRI outcomes represents a post hoc exploratory analysis.

## PATIENTS AND METHODS

Following research ethics approval (REC 07/Q0403/46) and informed parental consent, preterm infants admitted to the Chelsea and Westminster Hospital Neonatal Unit, London, UK (<32 weeks of gestation) and term infants (37–42 weeks gestation) on the postnatal ward were recruited (January 2007–July 2008). Infants with congenital anomalies were excluded.

### Preterm nutrition and growth

Preterm nutritional practice during the study period has been previously described.[4] In brief, enteral feeds were started on day 1 with either maternal expressed breast milk or donor expressed breast milk prior to consideration of the use of formula. Human milk fed infants received Nutriprem breast milk fortifier (Cow & Gate) once 150 mL/kg/day of feed volume was reached. Parenteral nutrition was started on the day of birth.

Weight and head circumference data for the preterm group are expressed as SD scores (SDS) at birth and at the time of MRI. Birth length was not routinely measured during the study period. Macronutrient and human milk intake was recorded between birth and $34^{+6}$ weeks postmenstrual age (PMA) using a nutritional data capture system designed in-house (Nutcracker, Imperial College, London, UK). Macronutrient data were expressed as the difference between recommended daily intake (RDI) from Tsang et al[1] and actual daily intake in grams or kilocalories/kilogram for the period of either early (first week of life) or total nutrition (birth and $34^{+6}$ PMA).[4] Human milk intake is expressed as mL/kg/day and as percentage of enteral feeds given as human milk.

### MRI

MRI was performed at the Robert Steiner Unit, Hammersmith Hospital shortly after discharge from the neonatal unit as previously described[4] using a Philips Achieva system (Best, the Netherlands).

### Total and regional AT

Imaging parameters are shown in table 1. Regional AT depots were classified as total superficial subcutaneous, total deep subcutaneous and total internal and subdivided into abdominal or non-abdominal (figures 1 and 2). AT volumes were calculated as previously described.[6] Non-AT mass was calculated by conversion of AT volume to AT mass (density of AT of $0.9 \, g/cm^3$) and subtraction of the result from weight at the time of MRI.

### Brain volume

T2-weighted images were acquired using a dynamic sequence of six separate loops of single shot images which were then registered and reconstructed to produce volumetric datasets to eliminate the effects of motion artefact.[11] Imaging parameters are shown in table 1. Images were corrected for radiofrequency inhomogeneity (http://mipav.cit.nih.gov). BET Brain extraction tool FSL V.3 (http://www.fmrib.ox.ac.uk/fsl/)[12] was then used to delete non-brain tissue and create binary brain masks representing intracranial volume.[13] A mask of ventricular and cerebrospinal fluid (CSF) spaces was created using the thresholding feature of Image J, a java-based image processing program.[14] Brain volume was calculated by subtraction of the volume of the ventricular and CSF masks from the volume of the intracranial mask using ImageJ (figure 3). Brain MRIs were reported by MR for clinical purposes and given a score (0–13) adapted from Dyet et al[15] so that, if necessary, pathological findings could be accounted for.

### Proximal CAVT measurement and analysis

An optimised neonatal three-dimensional time of flight MR angiography sequence was used to assess the anterior, middle and posterior cerebral arteries.[7] Imaging parameters are shown in table 1. Vessel tortuosity was assessed using a previously validated measurement, distance factor[16] (figure 4), and a CAVT score was determined as a global measure of tortuosity for each participant by calculating the mean of the anterior cerebral artery, middle cerebral artery and posterior cerebral artery distance factor.

### Illness severity

CRIB II Score was calculated in preterm infants.[17]

### Sample size calculation and analyses

Based on previous work,[6] we estimated that recruitment of 60 preterm and 60 term infants allowed detection of a 0.5 SD difference between the groups for AT volume (power 80%, significance 5%). As this was an exploratory hypothesis-generating prospective observational study and given the uncertainty as to what differences were of clinical importance, additional sample size calculations were not considered. Data were adjusted for relevant confounding variables where appropriate and are presented for each of the MR outcomes with a comparison of outcomes between the groups. Data were analysed by comparison of means and tested for normality. Parametric or non-parametric methods were then applied accordingly. Within the preterm group, Pearson correlation was used to investigate the relationship between nutrition and MR outcomes at term age and these were limited to the patients in whom both nutritional data and the MRI outcome were successfully acquired. For a multivariate analysis, a minimum of 10n participants is considered appropriate, where n is the

**Table 1** MRI parameters for adipose tissue, brain volume and proximal cerebral arterial vessel tortuosity

| | Planes | Weighting | FOV | Slices | Slice thickness (mm) | Slice gap (mm) | TE (ms) | TR (ms) | Acquisition time (s) |
|---|---|---|---|---|---|---|---|---|---|
| Adipose tissue[6] | Axial | T1 fast spin echo | 300 | 25 | 5 | 5 | 20 | 500 | 360 |
| Brain volume[11] | Axial/sagittal | T2 | 220 | 60 | 3 | −1.5 | 160 | 16646/38 662 | 425 |
| Proximal cerebral arterial vessel tortuosity[7] | Axial/sagittal | Angiogram | 175 | 100 | 0.6 | 0 | 8.06 | 19 | 329 |

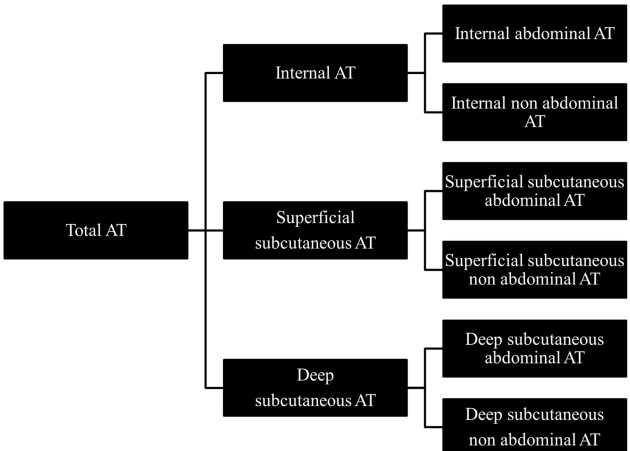

**Figure 1** Classification of adipose tissue depots into internal, superficial subcutaneous and deep subcutaneous depots and further subclassification according to abdominal or non-abdominal position.

number of covariates. Data are presented as mean (SD) or mean (95% CI).

## RESULTS

Sixty-one infants were recruited during the study (preterm 22, term 39). MR images without motion artefact were acquired as follows: AT volume (preterm 22, term 39), brain volume (preterm 19, term 19) and CAVT measurement (preterm 20, term 13). In preterm infants, mean (SD) CRIB II score was 7.6 (4.2). Twenty-five per cent had chronic lung disease of prematurity (defined as an oxygen requirement at 36 weeks PMA), 20% had a patent ductus arteriosus requiring pharmacological therapy, 5% had retinopathy of prematurity requiring laser therapy and 5% had a large intraventricular haemorrhage (grade III/IV

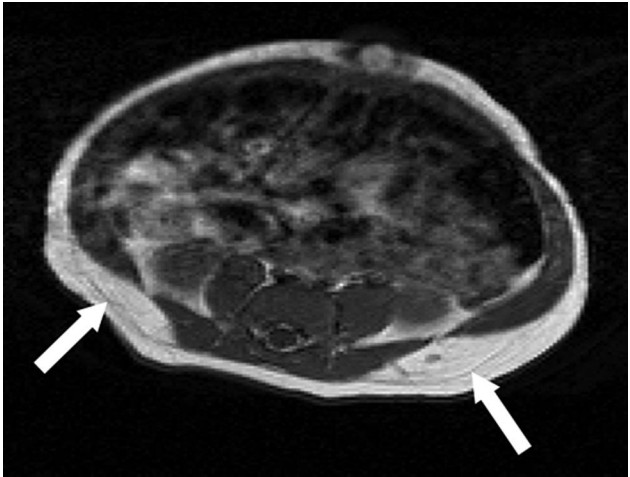

**Figure 2** T1-weighted axial MRI (abdominal level) demonstrating the deep and superficial subcutaneous adipose tissue depots. A clear fascial plane is noted between the superficial and deep subcutaneous layers (arrows).

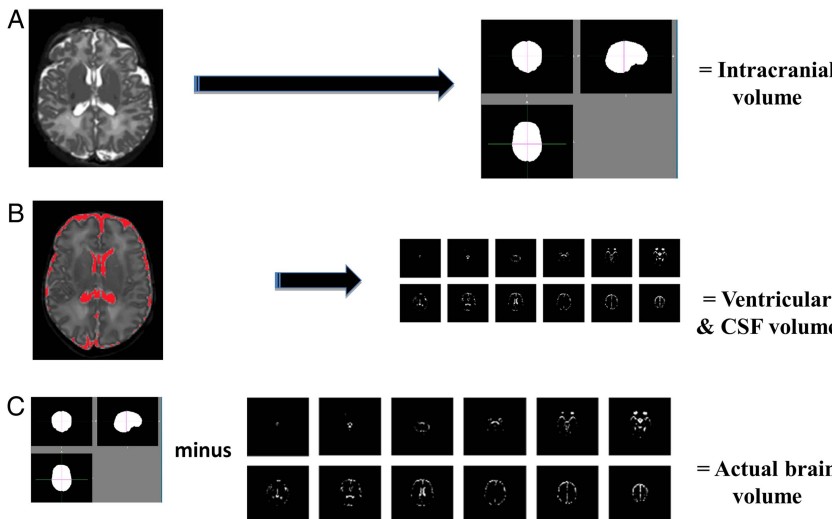

**Figure 3** (A) T2-weighted MRI undergoes RF inhomogeneity correction and subsequent creation of a binary brain mask of intracranial volume (BET brain extraction tool FSL V.3). (B) Creation of a mask of the ventricular and cerebrospinal fluid spaces using the thresholding feature of ImageJ V.1.38. (C) Calculation of actual brain volume by subtraction of ventricular and cerebrospinal volume from intracranial volume.

Papile classification). 0% had necrotising enterocolitis requiring surgery. Table 2 summarises the baseline and imaging characteristics and the MR outcomes of the preterm and term groups.

### Preterm nutrition and growth

Growth and nutrition data for the preterm cohort have been previously published.[4] In brief, the mean (SD) for birth weight and birth head circumference SDS were: −0.13 (0.78) and −0.64 (1.12). At the time of MRI, the respective values were −1.39 (0.93) and −0.86 (1.33). Weight gain was mean (SD) 9.48 (1.73) g/kg/day between birth and $34^{+6}$ weeks. Preterm macronutrient intakes for both the first week after birth and for the period from birth until $34^{+6}$ weeks PMA revealed a mean protein deficit (first week: −1.6 g/kg/day; birth until $34^{+6}$ weeks −0.4 g/kg/day)

in the context of excessive carbohydrate and fat intake. Mean (SD) human milk intakes for these periods were 36.6 (29.7) and 108.9 (46.6) mL/kg/day, respectively.

### Preterm nutrition and MRI outcomes
#### Human milk and MRI outcomes

After adjustment for weight at MRI, there was no correlation between early or total human milk intake and either regional AT volumes or brain volume at term age. There was no correlation between early human milk intake and overall CAVT score (r=0.31, p=0.18); however, there was a weak positive relationship between total human milk intake and overall CAVT score (r=0.44, p=0.05) (figure 5). This relationship was also apparent when total human milk intake was expressed as a percentage of total milk intake (r=0.45, p=0.04).

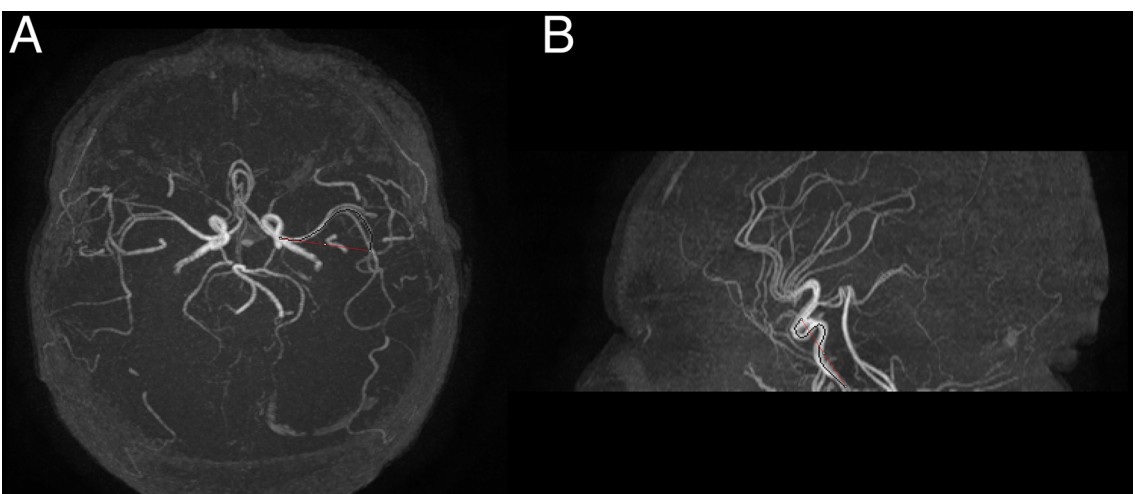

**Figure 4** (A) Axial MR angiogram demonstrating the tracing of a contour along the left middle cerebral artery for 30 mm (black line) along with a straight line measurement between the start point and the end point of the traced contour (red line). (B) Sagittal MR angiogram demonstrating the tracing of a contour along the anterior cerebral artery for 30 mm (black line) along with a straight line measurement between the start point and the end point of the traced contour (red line).

**Table 2** Birth characteristics, imaging characteristics and MRI outcomes in preterm-at-term and term infants

| | Preterm-at-term | n | Term | n | 95% CI for difference |
|---|---|---|---|---|---|
| **Birth characteristics** | | | | | |
| Gestation at birth (weeks) | 28.9 (2.8) | 22 | 40.2 (1.20) | 39 | N/A |
| Birth weight (kg) | 1.26 (0.41) | 22 | 3.40 (0.45) | 39 | N/A |
| Gestation at MRI (weeks) | 40.2 (3.10) | 22 | 41.4 (1.40) | 39 | N/A |
| **Anthropometry at the time of MRI** | | | | | |
| Weight at MRI (kg) | 3.01 (2.83–3.19) | 22 | 3.30 (3.17–3.44) | 39 | 0.30 (0.66 to 0.53)* |
| Length at MRI (cm) | 47.9 (46.9–48.9) | 22 | 51.3 (50.5–52.1) | 39 | 3.4 (2.1 to 4.7)** |
| Head circumference at MRI (cm) | 34.2 (33.6–34.7) | 22 | 35.2 (34.8–35.6) | 39 | 1.1 (0.4 to 1.7)** |
| **Somatic MRI outcomes** | | | | | |
| Total AT (L) | 0.782 (0.735–0.829) | 22 | 0.657 (0.623–692) | 39 | 0.125 (0.064 to 0.186)** |
| Total deep subcutaneous AT (L) | 0.033 (0.029–0.036) | 22 | 0.024 (0.021–0.027) | 39 | 0.009 (0.004 to 0.013)** |
| Total superficial subcutaneous AT (L) | 0.673 (0.631–0.715) | 22 | 0.568 (0.538–0.599) | 39 | 0.105 (0.051 to 0.159)** |
| Total internal AT (L) | 0.076 (0.069–0.083) | 22 | 0.065 (0.060–0.070) | 39 | 0.011 (0.002 to 0.020)* |
| Deep subcutaneous adipose abdominal AT (L) | 0.017 (0.014–0.019) | 22 | 0.012 (0.011–0.014) | 39 | 0.004 (0.001 to 0.005)** |
| Deep subcutaneous adipose non-abdominal AT (L) | 0.016 (0.014–0.018) | 22 | 0.012 (0.010–0.013) | 39 | 0.004 (0.002 to 0.007)** |
| Superficial subcutaneous adipose abdominal AT (L) | 0.133 (0.122–0.145) | 22 | 0.091 (0.083–0.100) | 39 | 0.042 (0.027 to 0.057)** |
| Superficial subcutaneous adipose non-abdominal AT (L) | 0.540 (0.506–0.574) | 22 | 0.477 (0.452–0.502) | 39 | 0.063 (0.019 to 0.107)** |
| Internal abdominal AT (L) | 0.022 (0.019–0.025) | 22 | 0.018 (0.016–0.021) | 39 | 0.004 (0 to 0.007) |
| Internal non abdominal AT (L) | 0.054 (0.049–0.060) | 22 | 0.047 (0.043–0.051) | 39 | 0.008 (0.001 to 0.005)* |
| Non-AT mass (kg) | 2.49 (2.45–2.54) | 22 | 2.61 (2.57–2.64) | 39 | 0.12 (0.06 to 0.170)** |
| **Brain MRI outcomes** | | | | | |
| Brain volume (mL) | 481.46 (462.29–498.63) | 19 | 474.02 (456.85–491.20) | 19 | 7.44 (18.53 to 33.40) |
| Cerebrospinal fluid volume (mL) | 63.78 (55.06–72.50) | 19 | 31.44 (22.72–40.16) | 19 | 32.34 (19.15 to 45.53)** |
| Anterior cerebral artery DF | 1.38 (1.30–1.46) | 20 | 1.25 (1.19–1.32) | 13 | 0.13 (0.03 to 0.23)* |
| Middle cerebral artery DF | 1.38 (1.32–1.44) | 20 | 1.26 (1.21–1.31) | 13 | 0.12 (0.04 to 0.19)** |
| Posterior cerebral artery DF | 1.46 (1.41–1.50) | 20 | 1.27 (1.24–1.30) | 13 | 0.19 (0.14 to 0.24)** |
| Proximal CAVT score | 1.41 (1.36–1.45) | 20 | 1.26 (1.23–1.29) | 13 | 0.14 (0.09 to 0.20)** |

Results are mean (SD) or mean (95% CI).
*p<0.05, **p<0.01.
AT, adipose tissue; CAVT, cerebral arterial vessel tortuosity; DF, distance factor; L, litres; n, sample size.

## Macronutrient intake and MRI outcomes

There were no correlations between macronutrient intake (protein, carbohydrate, fat) and total adiposity (n=22). After adjustment for PMA, neither early nor total preterm macronutrient intake was correlated with brain volume at term age (n=19). There were no relationships noted between early or total macronutrient intake and proximal CAVT score (n=20), a summary measure of tortuosity.

## Preterm at term versus term group comparison
### Anthropometry

Anthropometric data were adjusted for PMA at the time of MRI. Preterm infants were smaller, shorter and had smaller head circumferences than term infants (table 2).

### Adiposity

Adipose tissue data were adjusted for PMA at the time of MRI. Preterm infants had more AT than term infants

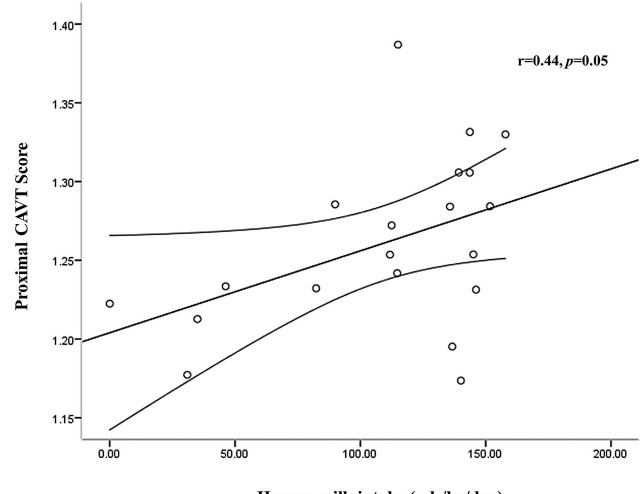

**Figure 5** Relationship between proximal cerebral arterial vessel tortuosity (CAVT) score at term and total human milk intake (birth to 34$^{+6}$ weeks PMA) in preterm infants.

with expansion of the superficial subcutaneous, deep subcutaneous and internal AT depots. There was a parallel reduction in non-AT mass (table 2).

## Brain volume

Brain MRI scores in the cohort of preterm infants were mean (SD) 3.6 (2.1). There was no correlation between brain MRI score and brain volume (r=0.31, p=0.20). At term age, preterm brain volumes were smaller than those for term infants (PT 461.74 (436.16–487.32) mL vs 493.74 (476.43–511.05) mL, p=0.05). However, after adjustment for weight at the time of MRI, a significant confounding factor, there was no difference between the groups. Preterm-at-term infants had significantly increased CSF volume (table 2).

## Distance factor and proximal CAVT score

In comparison to term born infants, preterm-at-term infants had a significant reduction in CAVT (table 2).

## Relationships between somatic and central nervous system MRI phenotype

After adjustment for weight at MRI, there was a significant negative correlation noted in preterm-at-term infants but not in term infants, between deep subcutaneous AT volume and brain volume (r=−0.58, p=0.01) (table 3). There were no statistically significant correlations between regional AT and CAVT score.

## DISCUSSION

We identify novel findings of interest including a positive association between human milk intake and proximal CAVT, a marker of cerebrovascular development and a negative correlation between regional adipose tissue volume and brain volume in preterm infants. The study also comprehensively characterises the somatic and brain phenotypes of a cohort of preterm infants at term in comparison to term born healthy infants and demonstrates (1) reduced anthropometric measures, (2) increased total and regional adiposity, (3) reduced non-adipose tissue mass, (4) comparable brain volumes, (5) increased CSF volume and (6) reduced proximal cerebral arterial vessel tortuosity. We have shown no

**Table 3** Pearson correlations between regional AT depots and brain volume in preterm-at-term and term infants adjusted for weight at imaging

| Regional AT depot | Preterm-at-term (n=19) | Term (n=19) |
|---|---|---|
| Total AT | −0.23, p=0.38 | −0.23, p=0.35 |
| Total DSC AT | −0.58, p=0.01* | −0.05, p=0.85 |
| Total SSC AT | −0.22, p=0.39 | −0.24, p=0.34 |
| Total I AT | −0.09, p=0.73 | −0.13, p=0.96 |

AT, total adipose tissue; IA AT, internal abdominal adipose tissue; Total DSC AT, total deep subcutaneous adipose tissue, Total I AT, total internal adipose tissue; Total SSC AT, total superficial subcutaneous adipose tissue.

relationship with either body composition or brain volume at term age within the range of macronutritional intakes received by the preterm infants in this study.

Key study strengths include the comprehensive ascertainment of preterm nutritional data and the assessment of a number of MR outcomes. Limitations include the prospective observational design of the study that preclude any inferences regarding causality, and the 'suboptimal' preterm nutritional intake, a potential determinant of the observed phenotype. Successful acquisition of a number of different MR outcomes without use of sedation was challenging and the motion artefact meant that not all recruited infants had data of sufficient quality for analysis. We were unable to recruit the desired number of infants and recognise that the study may be underpowered. The preterm infants studied were relatively healthy, as evidenced by low illness severity (CRIB II scores), low incidences of serious neonatal morbidity and low brain MR scores, factors that may have attenuated any associations with nutritional intake.

The finding of increased adiposity in preterm infants confirms our previous work.[6] We have previously shown expansion of the internal abdominal AT compartment in preterm-at-term infants in comparison to term born infants in a cohort of preterm infants recruited in 2002–2003,[6] a time at which neonatal nutrition was possibly not as carefully considered as it is today. This contrasts with our present data which demonstrate a global expansion of all AT depots. Whether the differences observed in AT partitioning between these studies relate to changes in nutritional practices and the provision of a more calorie dense diet is a plausible but as yet unproven hypothesis. Other potential mechanisms, which we have not explored, that might explain the increased adiposity seen in preterm infants include weight cycling and inflammation. It is known that cycling between high-calorie diets and low-calorie diets (weight cycling)[18] results in the preferential deposition of AT over non-AT mass[19] and this phenomenon often occurs in preterm infants when enteral feeds are discontinued because of concerns regarding feed intolerance and then restarted. Inflammation is also a known determinant of adiposity[20] and it is possible that the proinflammatory milieu often present in the perinatal period (maternal chorioamnionitis, use of intravenous lipid formulations high in omega-6 fatty acids and postnatal infection/ inflammation) may be relevant to the observed phenotype.

The nutritional intake received by this cohort of preterm infants was imbalanced (low in protein and high in fat and carbohydrate) and suboptimal in relation to expert consensus recommendations.[1] [2]) In animal models, protein deficiency is associated with a number of adverse health outcomes including a reduction in lifespan,[21] cardiovascular dysfunction,[22] reduced dendritic spine density[23] reduced brain weight[24] and reduced cortical blood vessel density[25] Human adult data indicate that dietary protein is an important factor in body

weight regulation.[26] Recent data in human ex-preterm infants suggest an association between early growth patterns and fractional anisotropy, a measure of brain microstructure.[27] Our group has previously shown that preterm-at-term AT deposition can be attenuated by use of fortified human milk.[28] Whether this translates into a longer term benefit is unknown.

The negative correlation we show between adiposity in preterm-at-term infants and brain volume is consistent with the findings in adults and children.[10 29 30] This, notwithstanding preterm-at-term brain volume, was comparable to that of term born healthy infants, which is also in keeping with previously published work.[31] This, together with the finding of maintained head circumference SDS between birth and time of imaging, may be indicative of 'brain sparing' in nutritionally compromised infants.

We have confirmed the finding of reduced proximal CAVT in preterm infants.[7] Though the natural history and long-term neurodevelopmental sequelae of reduced CAVT are unknown, epidemiological data indicate that advancing gestation confers a significant reduction in risk of fatal adult cerebrovascular disease (occlusive stroke).[32] Our observation that human milk may be protective despite low macronutrient density suggests that non-nutritive factors, such as vascular endothelial growth factor, may play an important role in cerebrovascular development.

In conclusion, we have extended the characterisation of the preterm-at-term phenotype. Our data do not support an association between macronutrient intake and body composition or brain volume. Other plausible determinants that remain to be explored are the roles of micronutrient deficiency, weight cycling, disease severity and chronic inflammation.

**Author affiliations**
[1]Section of Neonatal Medicine, Department of Medicine, Imperial College London, Chelsea & Westminster Hospital Campus, London, UK
[2]Department of Child Health, William Harvey Hospital, Ashford, Kent, UK
[3]Metabolic and Molecular Imaging Group, MRC Clinical Sciences Centre, Imperial College London, London, UK
[4]Division of Clinical Sciences, Imperial College London, MRC Clinical Sciences Centre Hammersmith Hospital, London, UK
[5]Department of Perinatal Imaging & Health, Division of Imaging Sciences & Biomedical Engineering, Faculty of Medicine, Centre for the Developing Brain, King's College London, St Thomas' Hospital, London, UK

**Contributors** VV helped design the original prospective study, analysed the brain MRI data, drafted the initial manuscript and approved the final manuscript as submitted. GD optimised the MRI parameters for the study. LT coordinated the analysis of the adipose tissue MRI data. CM optimised the MR angiography and advised on vessel tortuosity measurements. JB was involved in the study design and supervised the analysis of the adipose tissue MRI data. MAR was involved in the study design and supervised the analysis of the brain MRI data. NM was involved with the study design and supervised the analysis of neonatal nutritional data. All authors critically reviewed and revised the manuscript and approved the final manuscript as submitted.

**Funding** United Kingdom Medical Research Council; Chelsea and Westminster Hospital NHS Foundation Trust

**Competing interests** None.

**Provenance and peer review** Not commissioned; externally peer reviewed.

**Data sharing statement** No additional data are available.

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
