## [Reviewer comments · BMJ Open]

This paper was submitted to the ADC but declined for publication following peer review. The authors addressed the reviewers' comments and submitted the revised paper to BMJ Open. The paper was subsequently accepted for publication at BMJ Open.

ARTICLE DETAILS

TITLE (PROVISIONAL)	Preterm nutritional intake and magnetic resonance imaging phenotype at term age: A prospective observational study.
AUTHORS	Vasu, Vimal; Durighel, Giuliana; Thomas, E Louise; Malamateniou, Christina; Bell, Jimmy; Rutherford, Mary; Modi, Neena

VERSION 1 - REVIEW

REVIEWER	Dorling, Jon Queen's Medical Centre, Neonatal Unit
REVIEW RETURNED	10-Oct-2013

GENERAL COMMENTS	Minor suggestions;- Results 2nd sentence says CRIB II score was mean (SD) 7.6 (4.2), this would be better as "mean CRIB II score 7.6 (SD 4.2)". Figure 1a is difficult to read as it contains lots of abbreviations that are then expanded in the figure heading. Perhaps this could be done with the full text in the figure itself? Alternatively this could be explained alone in the text. In the discussion on p14 line 18, the statement "the wellknown macronutritional shortfalls of unfortified human milk, especially with respect to protein" should be backed up by suitable reference(s) Page 15 line 34, human milk intake exert a beneficial should contain 'exerts' instead of exert. The association of the total human milk intake with brain blood vessels changes is the main finding of this paper. It would strengthen the paper to review the previous evidence in more detail. It would be good to postulate more on the possible mechanisms for this effect also.
--

- This manuscript received two reviews at the ADC but the other referee had declined to make his reviews public.

VERSION 1 – AUTHOR RESPONSE

Minor suggestions;-

Results 2nd sentence says

CRIB II score was mean (SD) 7.6 (4.2), this would be better as "mean CRIB II score 7.6 (SD 4.2)".

The manuscript has been amended accordingly

Figure 1a is difficult to read as it contains lots of abbreviations that are then expanded in the figure heading. Perhaps this could be done with the full text in the figure itself? Alternatively this could be explained alone in the text.

We have amended Figure 1 to incorporate the Reviewers suggestion but in doing do have also created a separate Figure (Figure 2) to illustrate how superficial and deep AT depots can be differentiated using MR imaging.

In the discussion on p14 line 18, the statement "the wellknown macronutritional shortfalls of unfortified human milk, especially with respect to protein" should be backed up by suitable reference(s)

In re-writing the manuscript to incorporate the 2 submitted manuscripts we have removed this sentence and so there is no longer a requirement for a reference.

Page 15 line 34, human milk intake exert a beneficial should contain 'exerts' instead of exert.

The manuscript has been amended accordingly

The association of the total human milk intake with brain blood vessels changes is the main finding of this paper. It would strengthen the paper to review the previous evidence in more detail. It would be good to postulate more on the possible mechanisms for this effect also.

We agree with this comment and only limited the discussion in this area for sake of word limit.

The amended manuscript has been revised to elaborate on this area as follows:

"Though the natural history and long term neurodevelopmental sequelae of reduced CAVT are unknown, epidemiological data indicate that advancing gestation confers a significant reduction in risk of fatal adult cerebrovascular disease (occlusive stroke) (32). Our observation that human milk may be protective despite low macronutrient density, suggests that non-nutritive factors ,such as vascular endothelial growth factor may play an important role in cerebrovascular development".